

# Earth System Modelling on System-level Heterogeneous Architectures: EMAC (version 2.42) on the Dynamical Exascale Entry Platform (DEEP)

**M. Christou[1], T. Christoudias[1], J. Morillo[2], D. A. Mallon[3] and H. Merx[1, 4]**

[1]{The Cyprus Institute, Nicosia, Cyprus}

[2]{Barcelona Supercomputing Center, Barcelona, Spain}

[3]{Jülich Supercomputing Centre, Jülich, Germany}

[4]{Max Planck Institute for Chemistry, Mainz, Germany}

Correspondence to: T. Christoudias (christoudias@cyi.ac.cy)

**Abstract**

We examine an alternative approach to heterogeneous cluster-computing in the many-core era for Earth System models, using the European Centre for Medium-Range Weather Forecasts Hamburg (ECHAM)/Modular Earth Submodel System (MESSy) Atmospheric Chemistry (EMAC) model as a pilot application on the Dynamical Exascale Entry Platform (DEEP). A set of autonomous coprocessors interconnected together, called Booster, complements a conventional HPC Cluster and increases its compute performance, offering extra flexibility to expose multiple levels of parallelism and achieve better scalability. The EMAC model atmospheric chemistry code (Module Efficiently Calculating the Chemistry of the Atmosphere (MECCA)) was taskified with an offload mechanism implemented using OmpSs directives. The model was ported to the MareNostrum 3 supercomputer to allow testing with Intel Xeon Phi accelerators on a production-size machine. The changes proposed in this paper are expected to contribute to the eventual adoption of Cluster-Booster division and Many Integrated Core (MIC) accelerated architectures in presently available implementations of Earth System Models, towards exploiting the potential of a fully Exascale-capable platform.



## 1 Introduction

The ECHAM/MESSy Atmospheric Chemistry (EMAC) model is a numerical chemistry and climate simulation system that includes sub-models describing tropospheric and middle atmosphere processes and their interaction with oceans, land and human influences (Jöckel et al., 2010). It uses the second version of the Modular Earth Submodel System (MESSy2) to link multi-institutional computer codes. The core atmospheric model is the 5th generation European Centre for Medium Range Weather Forecasts Hamburg general circulation model (ECHAM5, Roeckner et al., 2003, 2006).

The EMAC model runs on several platforms, but it is currently unsuitable for massively parallel computers, due to its scalability limitations and large memory requirements per core. EMAC employs complex Earth-system simulations, coupling a global circulation model (GCM) with local physical and chemical models. The global meteorological processes are strongly coupled and have high communication demands while the local physical processes are inherently independent with high computation demands. This heterogeneity between different parts of the EMAC model poses a major challenge when running on homogeneous parallel supercomputers.

We test a new approach for a novel supercomputing architecture as proposed by the DEEP project (Eicker et al., 2013, 2015, Mallon et al., 2012, 2013, Suarez et al, 2011), an innovative European response to the Exascale challenge. Instead of adding accelerator cards to Cluster nodes, the DEEP project proposes to use a set of interconnected coprocessors working autonomously (called Booster), which complements a standard Cluster. Together with a software stack focused on meeting Exascale requirements—comprising adapted programming models, libraries and performance tools—the DEEP architecture enables unprecedented scalability. The system-level heterogeneity of DEEP, as opposed to the common node-level heterogeneity, allows users to run applications with kernels of high scalability alongside kernels of low scalability concurrently on different sides of the system, avoiding at the same time over and under-subscription.

The Cluster–Booster architecture is naturally suited to global atmospheric circulation–chemistry models, with global components running on the Cluster nodes exploiting the high-speed Xeon processors and local components running on the highly-parallel Xeon Phi co-processors. By balancing communication versus computation the DEEP concept provides a new degree of freedom allowing us to distribute the different components at their optimal parallelisation. The concept is depicted diagrammatically in Figure 1.



## 2 Overview of application structure

The EMAC model comprises two parts, the meteorological base model ECHAM, using a nonlocal, spectral algorithm with low scalability, and the modular framework MESSy, linking local physical and chemical processes to the base model, with high scalability. While the number of processors used for the base model is limited by the non-local spectral representation of global physical processes, local physical and chemical processes described by framework submodels run independently from their neighbours and present very high scalability.

### 2.1 Phases

The implementation of EMAC comprises two main phases, the base model ECHAM integrating the dynamical state of the atmosphere, and the MESSy framework that interfaces to $n$ submodels calculating physical and chemical processes. Among these submodels stands out the MECCA submodel (Sander et al., 2007). This submodel computes the chemical kinetics of the homogeneous gas-phase chemistry of the atmosphere, and deserves special mention due to its intrinsic parallelism, high computational demands, and load imbalance rising from its rigid coupling to the base model's parallel decomposition.

The ECHAM base model runs in parallel in the distributed-memory paradigm using the Message Passing Interface (MPI, Aoyama et al., 1999) library for communication; the MESSy framework inherits the parallel decomposition defined by the base model. While ECHAM has been shown to be able to exploit the shared-memory paradigm using the Open Multi-Processing (OpenMP) library (Dagum et al., 1998), no such effort had been undertaken for the MESSy model so far.

It is, however, currently not possible to delegate the whole MESSy subsystem to full multi-threaded execution as some physical processes are naturally modelled in a column-based approach, and are strongly dependent on the system states at their vertically adjacent grid points. The implementation of submodels simulating these processes consequently relies on the column structure inherited from the base model. Furthermore, even a coarser column-oriented multi-threaded approach is hindered by global-variable interdependencies between submodels.

Describing homogeneous gas phase chemical kinetics, the MESSy submodel MECCA executes independently of its physical neighbours and is not limited by vertical adjacency relations. As more than half of the total run-time is spent in MECCA for a typical model scenario, it seems adequate to concentrate on the MECCA kernel with strong algorithmic locality and small





communication volume per task. As sketched in Figure **1** the current implementation of
MECCA, developed in the DEEP project, is delegated to the Booster using a task-based
approach while both ECHAM and the remaining MESSy submodels are executed on the Cluster
in the distributed-memory paradigm.

## 2.2   Dominant factors

Implementing a spectral model of the dynamical state of the atmosphere, the ECHAM phase
comprises six transform and six transposition operations in each time step, as seen in Figure 3.
The data in memory for each time step (data size scales with the square of the model resolution)
is transposed in an all-to-all communication pattern, and this phase is dominated by network
bandwidth.
Figure 4 displays one time step traced with Extrae/Paraver (Extrae 2015, Paraver 2015) starting
with the end of the grid point calculations of the last time step—in which most processors are
already idle (orange) due to load imbalance and waiting for process 14 (blue) to finish running.
This is followed by the transpositions and Fourier and Legendre transformations (magenta),
which execute simultaneously as further analysis showed. After the transpositions a short
interval with all processors running (blue) can be identified with the time step integration in
spectral space, followed by the inverse transformations and transpositions and transport
calculations in ECHAM.
While the pattern described so far repeats towards the end of the displayed interval, the major
fraction of the time step is spent without communication, running (blue) or waiting (orange) in
calculations in MESSy in grid space. The MESSy phase comprises some 30 submodels that are
tightly coupled by exchanging the atmospheric observables using global variables. Model
performance depends largely on a virtual longitude run-time parameter exploiting cache line
adjacency of the grid point variables. Investigations during the first phase of the project
determined the load imbalance visible in Figure 4 to be caused by chemical processes computed
in the MECCA submodel.
The observed load imbalance is one of the main factors determining application scalability. It
is caused by an adaptive time-step integrator solving a system of differential equations. As the
stiffness of these equations representing homogeneous photochemical reactions varies by up to
two orders of magnitude due to changes in the intensity of sunlight, the adaptive integrator
demands varying amounts of run time accordingly (described in more detail in Section 2.3).



In the MECCA phase the algorithmically complex adaptive time-step differential equation
integrator operates on chemical concentrations of a total data size of the order of a few kilobytes
per grid point. Yet, as seen in Figure 5 that highlights the load imbalance caused by MECCA
and observed using Scalasca (Scalasca 2015), this phase consumes the major proportion of the
total execution time, it is compute-bound and an obvious candidate for offloading to
accelerators. It should be noted though, that in a regular architecture, accelerating this highly
parallel phase will not eliminate the load imbalance.

## 2.3  Scalability considerations

To test the model scalability out-of-the-box, the EMAC application has been ported to the
JUDGE cluster at JSC, and a representative benchmark with a horizontal resolution of 128 grid
points in longitudinal and 64 grid points in latitudinal direction with 90 vertical levels and a
spin-up period of 8 simulated months has been compiled, frozen and packaged to be used for
measurements. Table **1** details the experimental setup for the results shown in this section.
EMAC was benchmarked with different numbers of processors on JUDGE in order to
determine the run time behaviour of the total application. As shown in Figure 6 the application
scales up to 384 processes (16 nodes x 24 MPI processes each), at higher numbers the
performance decreases. Parallel execution speed is determined by the balance of three factors:
computation, communication, and load imbalance. The benchmarking setup for the JUDGE
cluster can be seen in Table 2. While the computational resources increase with additional
processors and therefore increase the application performance, communication demands
diminish the positive effect of the additional processors. Additionally, increasing the granularity
of the total workload also increases the load imbalance.
While the number of processors used for the distributed-memory part of the code is limited by
the scalability of the non-local representation of global physical processes in ECHAM, the local
processes in MESSy running independently from their neighbours scale very well. The MESSy
subsystem has not been designed for multi-threaded execution, though, and contains non-local
code due to characteristics of the physical processes and algorithmic design decisions. Some
physical processes are naturally modelled in a column-based approach, because they are
strongly dependent on the system states at vertically adjacent grid points, e.g. sunlight intensity
at lower grid points depending on the absorption at higher grid points, and precipitation



depending on the flux of moisture from vertically adjacent grid cells. Sub-models simulating
these processes consequently rely on the column structure implemented in the current model.
In the existing distributed-memory parallel decomposition, the three-dimensional model grid is
split horizontally using two run-time parameters, setting the number of processes in latitudinal
and longitudinal direction. As work is distributed independently for each direction, a
rectangular decomposition is obtained.
The physical load-imbalance, caused by photo-chemical processes in the lower stratosphere and
natural and anthropogenic emissions, appears in the run time spent for each grid point when
examining the benchmark calculations. In Figure 7 the maximal MECCA kernel execution
wall-time for one grid point in each column differs by up to a factor of four. The load imbalance
is caused by the adaptive time-step integrator solving the differential equations that describe
the chemical equations computed in the MECCA submodel. The strongly varying light intensity
at sunrise and sunset and night-time emissions lead to stiff differential equations that require
more intermediate time steps with derivative function evaluations and increase the
computational load by up to one order of magnitude.
At high levels of parallelisation, the load imbalance becomes a limiting factor, and the factors
determining scalability in absolute numbers in Figure 8 are both communication and
computation. For the ECHAM phase (blue), when scaling to beyond 8 nodes the
communication demands of the underlying spectral model involving several all-to-all
communication patterns start to dominate.
In Figure 9 the point at which communication and computation require equal times around 8
nodes is clearly apparent; 16 nodes is commonly used in production runs of the EMAC
atmospheric model as a scientific application to balance efficiency and total required wall time.
**3   Model Developments**
**3.1   Intranode taskification**
The EMAC model atmospheric chemistry code (MECCA) was taskified using OmpSs (Bueno,
J. et al., 2011, 2012, Duran, A. et al., 2011, Florentino et al., 2014) directives. OmpSs allows
the user to specify inputs and outputs for blocks of code or functions, giving enough information
to the runtime to construct a dependency graph. This dependency graph reflects at all moments
which tasks are ready to be executed concurrently, and therefore the programmer does not have



to explicitly manage the parallelisation. This idea of tasks and task dependencies has been
adopted in the OpenMP 4.0 standard (OPENMP 4.0, 2013). Since in MECCA each gridpoint is
completely independent of its neighbours, this part of the code is in principle embarrassingly
parallel, with no communication or inter-task dependencies involved.
The MECCA submodel was refactored through the creation of computational kernels for
intranode parallelisation with shared-memory tasks. The new version of EMAC, running
ECHAM with MPI processes and MECCA with shared-memory OmpSs tasks outperforms the
old EMAC using pure MPI, and continues to scale beyond the region where the original
implementation scaling performance plateaus. This can be seen in Figure 10, which shows the
performance using multi-threading on the DEEP Cluster.
**3.2   Internode taskification**
In DEEP, OmpSs has been extended to support offloading tasks to remote nodes (Beltran et al.,
2015). This mimics the behaviour of other accelerator APIs that move data from the host to the
device, compute in the device, and return the results to the host. However, OmpSs adds two
very important features: i) it allows offloading to remote nodes, not just locally available
coprocessors/accelerators, which is a key functionality to effectively use the Booster; and ii) it
allows using the Booster as a pool of coprocessors, so tasks can be offloaded to any Booster
node with enough free cores. The latter enables to eliminate the load-imbalance caused by
sunlight gradients in MECCA.
In a shared-memory taskification the data is already shared between threads, and no memory
copies are necessary. However, in DEEP, to leverage the Booster, this data has to be copied to
the Booster nodes. Keeping that in mind, the new task-based MECCA implementation was
optimised and the memory and network footprint of the distributed-memory offloading was
reduced by three orders of magnitude. To minimise the memory footprint for offloaded tasks,
the number of computational grid elements issued to MESSy is further split into individual
elements for each task, by rearranging the grid point arrays in each time step to implement data
locality at the grid-point level, resulting in a reduction of the total memory footprint from 2.7
MB down to 6.3 KB for each task. This was the result of refactoring both the data and code
structures in MECCA. At the benchmark resolution of T42L90MA a total number of 737 280
tasks are generated in each time.



As discussed in section 2.2, a detailed analysis of the EMAC run-time behaviour using Scalasca (Wolf et al., 2008) and Extrae/Paraver has identified that the MECCA submodel consumes a major proportion of the execution time, does not participate in communication, and is independent of adjacency constraints. It is thus well suited to be delegated to the Booster employing the large dynamical pool of accelerator resources provided by the DEEP concept for load balancing of the heavily varying computation demands discussed in section 2.3.

Additionally, the distributed-memory offloading code was redesigned to exploit shared memory within the Xeon Phi many-core processors by nesting an OmpSs shared-memory region within Cluster-to-Booster tasks encompassing variable, runtime-defined number of individual gridpoint calculations. Thus, the number of tasks to be sent to the Booster can be controlled and optimised for each architecture, and host-specific configuration allows for optimum task size based on bandwidth, reducing task communication overheads.

With this approach, the specifics of the DEEP system architecture, and in particular the hardware present in MIC coprocessors is exploited by massively parallelising the chemistry calculations at the gridpoint level and offloading to the Booster exposing a significant amount of thread parallelism. At the same time the load imbalance observed in MECCA is automatically alleviated through OmpSs' dynamic load balancing by selecting a sufficiently fine task size and decoupling the model-domain location of the grid point from the task execution on the physical CPU.

### 3.3 Vectorisation

The computational core of MECCA is connected by an interface layer to the MESSy framework, integrating different submodel code and data structures into the ECHAM base model. It provides the gridpoint data as sub-arrays of the global simulation data – which have been rearranged from their native longitude and latitude coordinates into a virtual longitude and an outer index variable counting the virtual longitude blocks and assuming the role of a virtual latitude. The virtual longitude exploits cache line adjacency on non-vector architectures and serves as run-time vectorisation parameter for all MESSy submodels.

For the MECCA submodel an integrator kernel has been created that can be offloaded onto worker threads running on the main processor or hardware accelerators. The chemical mechanism is compiled by the Kinetic Pre-processor (KPP, Damian et al., 2002) implementing a domain-specific language for chemical kinetics. The integrator kernel operates on the



variables of one grid-point describing the local state of the atmosphere and the integrator

parameters determining the solution of the chemical equations. As the kernel variables are

passed as one-dimensional sub-arrays of global, four-dimensional arrays, extending along the

virtual longitude, the vector variables are transposed to extend contiguously along the

dimension of chemical species.

In order to estimate the run-time effect of the changes in the code, the application was

benchmarked on the DEEP Cluster using the Xeon main processors without vectorisation as

baseline measurement of 548.65 seconds per simulated day. Compiling with the auto-vectoriser

enabled for the AVX instruction set extensions decreased the run time to 466.40 seconds,

resulting in a first speed-up of 1.18. Examination of the optimisation report identified several

unaligned array accesses, which were solved using compiler directives and introducing aligned

leading dimensions at 64-byte boundaries for multi-dimensional arrays as needed for the

instruction set of Intel Xeon Phi. These changes improved the total application performance to

349.50 seconds per simulated day for a second speed-up of 1.33 achieving a total speed-up of

1.57 (Figure 11).

## 4    Attainable Performance

At the time of writing this manuscript the DEEP Booster is in the bring-up phase, and not

available to users. In order to project the performance of the full DEEP System, Xeon-based

measurements on the DEEP Cluster were combined with Xeon Phi-based measurements on

MareNostrum 3. The DEEP Cluster reference data weighted by the relative factors for each

phase derived from the metrics measurements exhibit a performance maximum for the base

model (ECHAM) and MESSy (excluding MECCA) at 8 nodes, representing a good estimate

for the optimal parallelisation of that phase on the Cluster. This estimate of 375 s per simulated

day for the low-scaling Cluster phases was used to extrapolate the attainable performance,

merging this result at 8 nodes, with the Xeon Phi data retrieved from MareNostrum 3, where

benchmarks using one node had been run with varying numbers of processing elements within

one Xeon Phi processor. The pure MPI time in the DEEP Cluster, the time for each phase, and

the theoretical performance when offloading to the Booster are shown in Figure 12.

While the number of Booster nodes required to attain similar performance to the original

distributed-memory based implementation corresponds to regular accelerator architectures with

individual boosters directly attached to cluster nodes, the projected DEEP performance scales



beyond the optimal performance achieved so far. The EMAC atmospheric chemistry global climate model seems therefore well suited to exploit an architecture providing considerable more hardware acceleration than provided by regular systems. The projected attainable performance that outperforms the pure-MPI conventional cluster paradigm at higher core count (depicted here as the number of Booster nodes, while keeping the ECHAM/MESSy MPI part on 8 Cluster nodes for optimal performance) is also shown in Figure **12**.

## 5  Conclusions

The global climate model ECHAM/MESSy Atmospheric Chemistry (EMAC) is used to study climate change and air quality scenarios. The EMAC model is constituted by a nonlocal meteorological part with low scalability, and local physical/chemical processes with high scalability. The model's structure naturally suits the DEEP Architecture using the Cluster nodes for the nonlocal part and the Booster nodes for the local processes. Different implementations of the code's memory and workload divisions were developed and benchmarked to test different aspects of the achievable performance on the proposed architecture. The use of the OmpSs API largely frees the programmers of implementing the offloading logic and, given that EMAC is developed and used in a large community working on all aspects of the model, can facilitate adoption of the concept in the MESSy community.

The chemistry mechanism was taskified at the individual gridpoint level using OmpSs directives. The chemistry code was refactored to allow for memory adjacency of vector elements. Enabling the vectoriser achieves a total speed-up of 1.57 by aligning all arrays at 64-byte boundaries. The OmpSs taskification with remote offload allows for massive task parallelisation and the implementation of optional two-stage offload to control Cluster-Booster task memory size and optimum bandwidth utilisation.

The computational load imbalance arising from a photochemical imbalance is alleviated at moderate parallelisation by assigning grid points with differing run times to each process and distributing the load over all processes. Due to the physical distribution of sunlight this load balancing does not require an explicit algorithm at moderate parallelisation; instead, the implicit assignment of the model grid in rectangular blocks suffices for this purpose. At higher numbers of processors this implicit load-balancing decreases and the resulting load imbalance has to be solved by active balancing. The dynamic scheduling provided by the OmpSs run-time system balances the computational load without a possible, but expensive prediction for the current time step.



With these approaches, the specifics of the DEEP System architecture, and in particular the hardware present in MIC coprocessors can be exploited by massively parallelising the chemistry calculations at the gridpoint level and offloading to the Booster exposing a significant amount of thread parallelism. At the same time the load imbalance observed in MECCA will be automatically alleviated through dynamic load balancing by selecting a sufficiently fine task size and decoupling the model-domain location of the cell from the task execution on the physical CPU.

Benchmark projections based on available hardware running the DEEP software stack suggest that the EMAC model requires the large numbers of Xeon Phi accelerators available in the DEEP architecture to scale beyond the current optimal performance point and exploit Amdahl's law with the highly scalable gridpoint calculations while capitalising on the high performance and fast communication for the spectral base model on Intel Xeon processors.

The changes proposed in this paper are expected to contribute to the eventual adoption of MIC accelerated architectures for production runs, in presently available implementations of Earth System Models, towards exploiting the potential of a fully Exascale-capable platform.

**Code Availability**

The Modular Earth Submodel System (MESSy) is continuously further developed and applied by a consortium of institutions. The usage of MESSy and access to the source code is licenced to all affiliates of institutions which are members of the MESSy Consortium. Institutions can become a member of the MESSy Consortium by signing the MESSy Memorandum of Understanding. More information can be found on the MESSy Consortium Website (http://www.messy-interface.org).

**Acknowledgements**

The research leading to these results has received funding from the European Community's Seventh Framework Programme (FP7/2007-2013) under Grant Agreement n° 287530.



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



1    Table 1: Experimental setup for JUDGE scalability test out-of-the-box.

| Scaling | Number of columns | Number of grid points | Number of chemical species | Spectral resolution |
|---|---|---|---|---|
| Strong scaling | 8192 columns with 90 levels | 737280 grid points | 139 species in 318 reactions | T42L90MA with 42 coefficients |





1    Table 2: System setup details for the analysis done on the JUDGE system.

| Backend compiler version | MPI runtime version | Compilation flags | MPI processes per node |
|---|---|---|---|
| Intel 13.1.3 | Parastation/Intel MPI 5.0.27 | -O3 –fp-model source –r8 –align all | 24 |





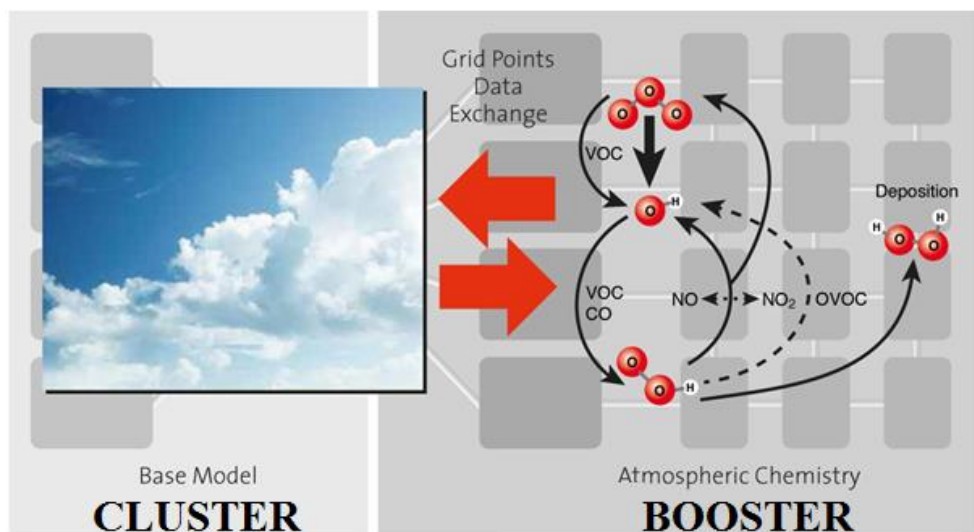

2    Figure 1 : Distribution of the Earth System Model components on the Cluster-Booster architecture.



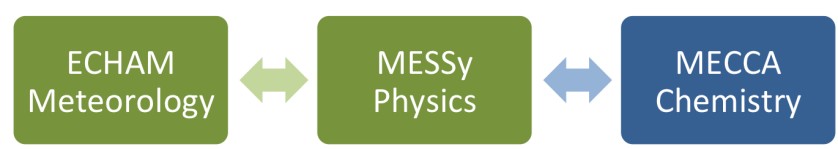

2 Figure 2 Phases of EMAC. Green phases run on the Cluster, blue phases run on the Booster.



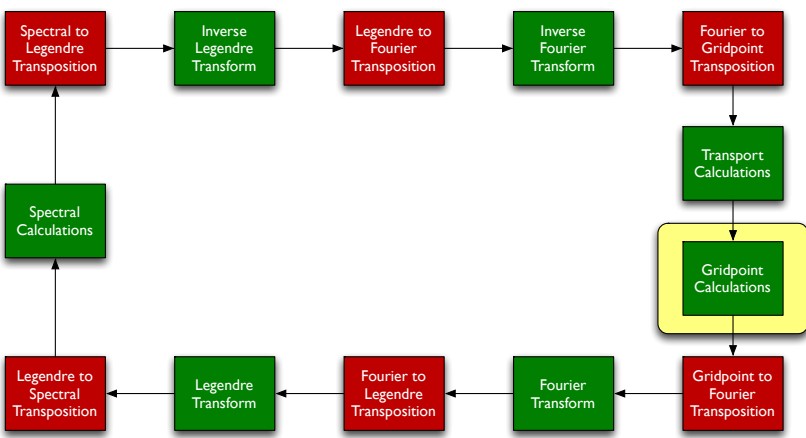

2    Figure 3 : The ECHAM main application loop. MESSy replaces and enhances the grid point calculations marked

3    in yellow.



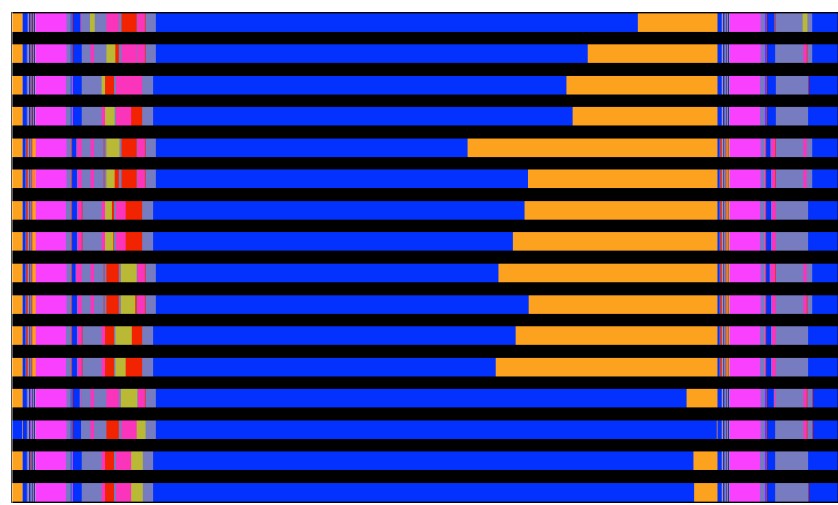

2  Figure 4 : Paraver trace of major processor usage of one time step. Time is along the horizontal and each bar

3  corresponds to a separate CPU core. Blue colour depicts computation; orange corresponds to idle time due to load

4  imbalance. The grid-space transpositions and Fourier and Legendre transformations are shown in magenta.





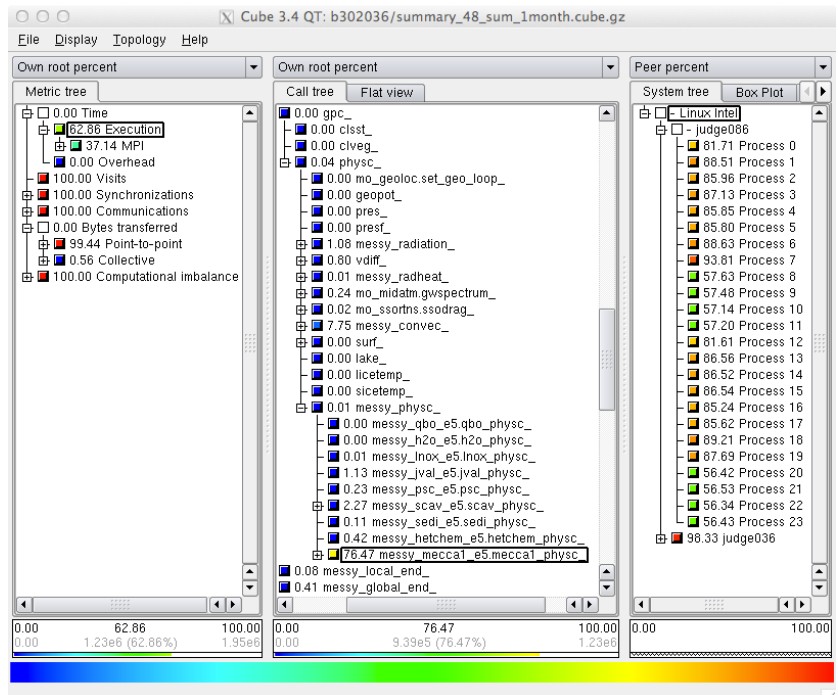

2    Figure 5 : MECCA execution time analysed with Scalasca.





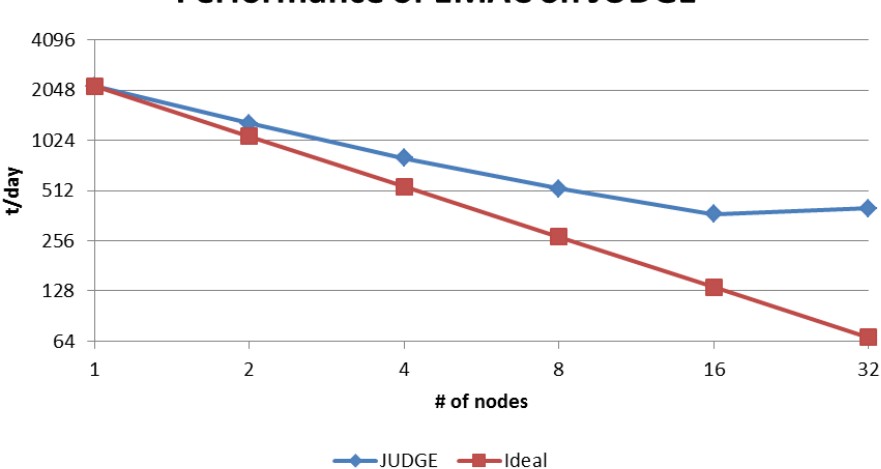

2    Figure 6 : Wall time for one simulated day versus the number of nodes on JUDGE.





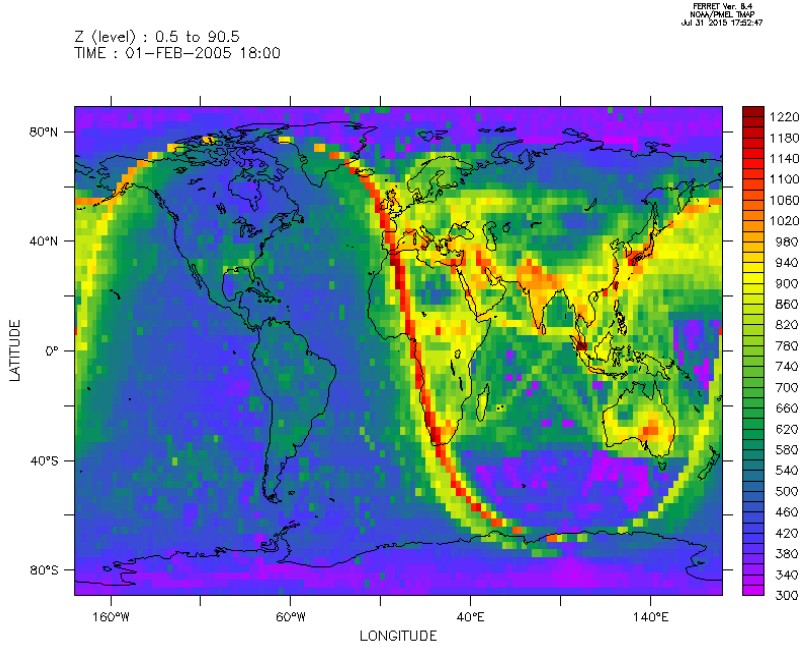

2 Figure 7 : Maximal MECCA kernel execution wall-time in microseconds. The adaptive time-step integrator shows

3 a non-uniform run time caused by stratospheric photochemistry and natural and anthropogenic emissions.





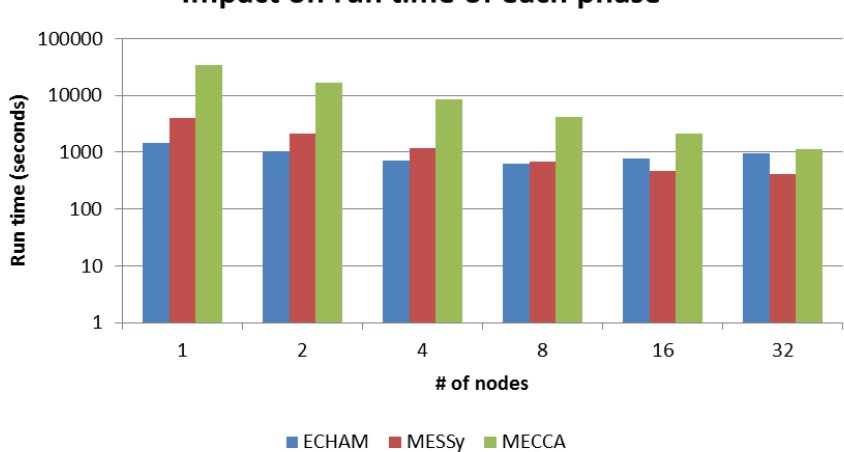

2    Figure 8 : Impact on run time of each phase of EMAC, when running on MareNostrum 3.



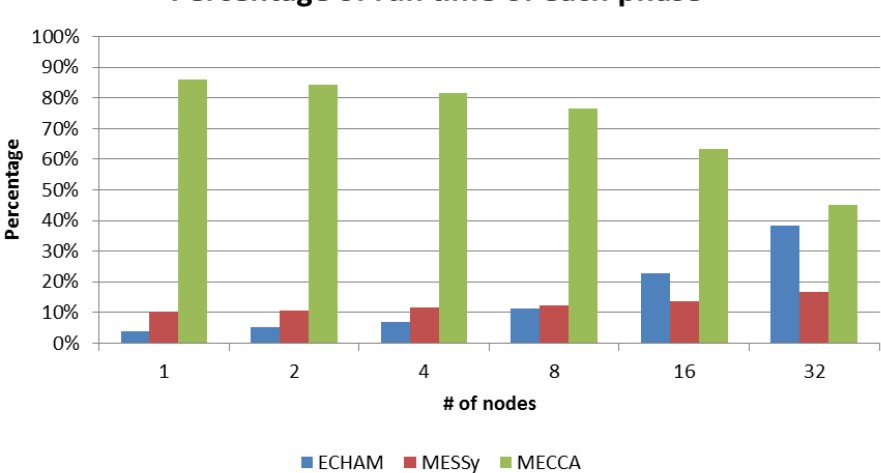

2    Figure 9 : Percentage of run time of each phase of EMAC, when running on MareNostrum 3.





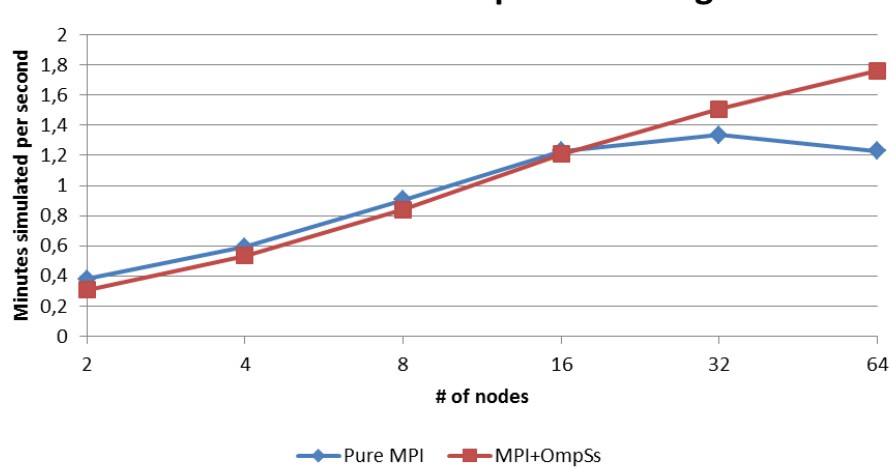

2    Figure 10 : Performance of OmpSs threading in the DEEP Cluster.



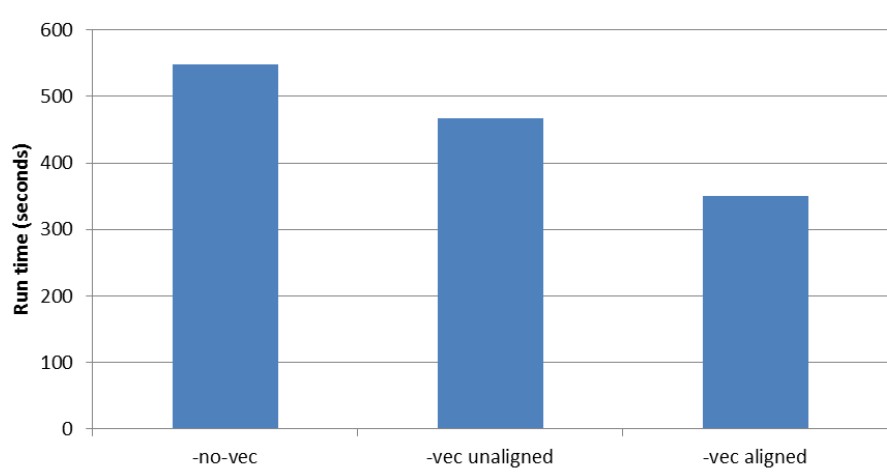

2    Figure 11 : Performance of the vectorisation of MESSY in a Xeon E5-2680.



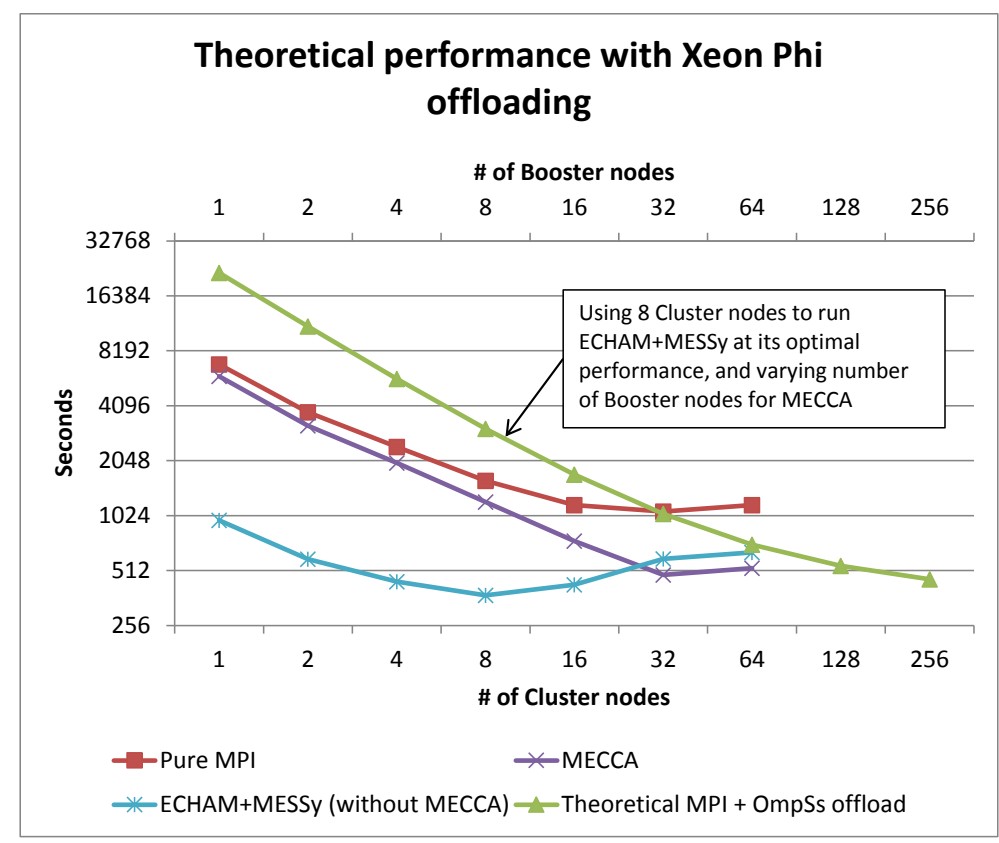

2    Figure 12: Time per simulated day in DEEP using a pure MPI approach, and a theoretical performance with

3    offloading to Xeon Phi, based on the metrics collected in MareNostrum 3. The theoretical MPI + OmpSs offload

4    data is based on a fixed configuration on the Cluster using 8 nodes, and scaling the number of Booster nodes.