# Peer review of "Earth System Modelling on System-level Heterogeneous Architectures: EMAC (version 2.42) on the Dynamical Exascale Entry Platform (DEEP)"

_Geoscientific Model Development, 2015_

## Referee Comment (RC1) · Anonymous Referee #1 · 12 Apr 2016

The article reports about one part of the DEEP project examining the approach of heterogeneous cluster-computing for Earth System models. In particular, this article is about the GCCM EMAC. A performance analysis shows, that the chemistry submodel MECCA is the bottleneck of usual EMAC simulations. As MECCA is solving the chemistry within each grid box independently, it is embarrassing parallel and thus the ideal candidate for application on a Booster architecture.

The article addresses a very important issue of Earth system modelling. The fact that these codes have a long history brings about that they are not at all optimised for modern computer architectures. Thus the refactoring of the code, to used the possibilities provided by current and future computing architectures is a very important issue.

[Figure]

Nevertheless, there are some issues that need to be improved upon revision. First of all the specific model configuration used for the scaling tests is not provided. But the setup heavily influences the performance of the model. Additionally, the explanations provided in this article are only correct for certain model setups. Therefore the setup needs to be provided. Secondly, this is a GMD article and the authors classified it "Development and technical paper". Thus I expect the authors to provide much more details about the developments themselves. How exactly have they been implemented? In general, the performance analysis is as long as the description of the developments. The latter should be the main part of the article from my point of view.

Below I give the details about the points raised above and name some additional issues. In summary, I am very much in favour of publishing this article. Nevertheless, major revisions are required.

Specific Comments

1. The authors often use the term "meteorological model" when they mean the dynamical core of the model. Meteorology comprises the dynamical processes as well as the physical processes such as cloud and precipitation formation or radiation. EMAC uses the dynamical core of the ECHAM model, but all physical processes are – by now – modularised as MESSy submodels. In ECHAM the dynamical core operates in spectral space, the physical processes are implemented in grid point space. All MESSy submodels (so far) are implemented as additions in the grid point space. Therefore the authors should be consistent in the terms they use throughout the article. Please change accordingly:

   - page 2 line 12
   - p.3, l. 2 + l. 6
   - p.5 l.24
   - p. 10 l.10

- Figure 2, left box

This list is not complete. So please check throughout the article.

2. To really understand the performance analysis, the full EMAC setup should be listed somewhere. If an ESCiMo-Setup or another published setup was used, this could simply be cited. If not, I would prefer to have a description, containing all details relevant for the publication, in the appendix. A zip-file containing the full namelist setup should be provided in the supplement.

3. Connected to the previous point: You characterize MECCA as the submodel computing "the chemical kinetics of the homogeneous gas-phase chemistry of the atmosphere," ... (e.g., p.3 l.13; p.4 l.29)

   In all MESSy setups which are not only focussed on the lower troposphere, heterogeneous processes on ice cloud are included via reaction rates provided by MSBM. As I do not know the specific setup of this study, I can not judge, if this statement is correct with repsect to it. As a general statement it is definitely not correct.

4. If MECCA is employed with a pure gas phase mechanism in this test setup, what will be the result, if heterogeneous reactions are included? This would establish a second source of imbalance caused by the appearance of PSCs.

5. SCAV uses KPP as well. In maximum it is called four times during one time step (for grid/subgrid scale liquid/ice clouds). Here the load imbalance is caused by the distribution of clouds over the model domain and additionally, SCAV is column bound. At least in the conclusion or outlook I'd like to see a statement, how easily your developements could be applied to SCAV and (maybe) if you expect performance gains for SCAV as well.

6. Chapter 3: Model developments
   What I am really missing in this chapter is the "development" i.e., an explicit

mentioning of the code changes required. You describe them superficially with some words, what - at least for a MESSy developer / user - is really interesting is what the code changes look like. For which of the described changes did you change which code parts? Do they require changes in the MESSy submodel interface layer (SMIL) or in the core layer (SMCL). Here it is most interesting, if these developments require a change in the automatically generated code. If yes, MECCA knows two stages of "automation". First KPP produces the code automatically from the equation file. Secondly, KP4 can be applied in order to remove the indirect indexing and expand the original KPP code by an additional dimension to enable a better performance due to better cache usage. If there are changes in the automatically generated code parts, did you change the scripts performing the automation or did you just chance one MECCA setup (and running xmecca once would destroy all you efforts)?.

I assume that this information is only of interest for MESSy or MECCA developers / users and not to the general readership. Therefore I recomment to add the most important information to the article itself and to add a supplement describing the changes in more detail and providing information about how to use it (something like a user manual), which also is in accordance to the GMD guidelines.

Additionally, a description, how the optimise the setup as described in p.8 ll.10-12 should be provided.

7. p. 8 ll. 1-19: It is not clear to me, what these paragraphs are about: ll.1-6 are a repetition of what was written earlier. ll. 7-12: claim that the code can be optimised tailormade for each architecture but does not tell how. ll.13-19 simply state that these changes increase the performance, but does not show any proof. Somehow I miss the point here...

8. Section 3.3
In this section the authors must be much more precise. It is not clear, what exactly

[Figure]

Interactive
comment

the authors are discussing here. Principially, the ECHAM (and thus the MESSy) grid point code is decomposed in a way of artificial latitude bands. ECHAM provides a so-called "local loop": here a loop over the second horizontal dimension is established, reducing the size of the fields forwarded to the individual submodels within the local loop by one horizontal dimension. The length of the remaining horizontal dimension is called vector length and can be chosen by namelist for optimisation on different computing achitectures.

Thus MECCA, which is called in the local loop is called with this one horizontal dimension only. For a better performance the original KPP output can be expanded by this additional horizontal dimension times the vertical dimension. This code is automatically produced by calling KP4. It is not clear to me, about which of these different aspects the authors talk exactly. So please clarify this issue.

9. Code Availability: You state the general terms for the MESSy code. Nevertheless, you are presenting new code developments, therefore it would be good to know, if it will become part of the official MESSy version soon and if it is possible to get hold of the code prior to this.

Minor issues

- paragraph p.3 ll.22-27, it would be good to have examples here. You provide them on page 5 last two lines, but they are already here useful to understand which kind of processes you are talking about.

- I have mixed feelings about Fig. 1. It looks nice, but is it really required? Additionally, as processes as deposition occur on the right hand side, this figure is in contradiction to Fig. 2.

- p.4, l.1: Shouldn't it be Figure 2?

- p.4, ll.8-10: it scales with the square of the horizontal resolution.

- p.5 l.13/ Table 1: Table 1 does not give a clue about the model setup, it only contains different numbers calculated from the resolution of the model. In my opinion the first column does not contain any information I would expect in a table. That the numbers indicate the possibility for strong scaling should be stated in the text not in the table itself. What is meant by "42 coefficients"? Which coefficients? Do you mean because of T42? Than the reader anyhow understands what T42 refers to, or he/she does not become any wiser by reading "42 coefficients" (my opinion).

  The table would be much better readable, if columns and rows would be switched

- Table 2: Here the last statement for table 1 applies even more: the table is much better readable, if switch columns and rows are switched.

- p.6 ll. 3-6: This is not fully correct. From your description I visualise a decomposition where the domain is split up in rectangular grid boxes. But ECHAM is using a decomposition where each task gets two (independent) latitudinal bands (of arbitrary length in the longitudinal range). These bands are usually not even adjacent to each other.

- p.6 l. 8: add location of natural and anthropogenic emissions.

- p.6 ll. 16-19: Why are you only describing the results for ECHAM seen in Fig. 8? Add something about "computing time for MESSy is still decreasing" and "computing time for MECCA decreases stronger than MESSy". Additionally, please point to the logarithmic scale and to the fact, that "MESSy" means without MECCA.

- p.6 ll.21-23: I am missing a conclusion, you just describe the plot.

- p.7 last line: "each time step", "step" is missing.

- p.8 l.31: What is meant by "domain-specific language" ?

- p.9 l.29 ; It would be helpful for the reader if you could mention the colours of the respective lines in the text. Do I assume right that "MPI" is the sum of ECHAM+MESSy and MECCA? First I thought it is the MPI communiction time. Please try to clarify your description.

- Fig. 1: For me this figure produces more questions instead of assisting in understand the distribution on the two different computing architecture parts. The "base model cluster" part contains a picture of a cloud, i.e., physical processes, and the "atmospheric chemistry booster" part does not only contain the chemical mechanism, but also deposition, thus it is not quite clear where the separation between cluster and booster should appear.

- Fig. 3: It is not clear on which ground the colours of the boxes are chosen. Personally I think, the figure overemphasises the dynamical core (including the transformations from grid point to spectral and vice versa). Because the grid point calculations contain much more sub processes which are completely left out by this figure.

- Fig. 5: Please provide a more descriptive caption for this figure. Not every reader is familiar with Scalasca output.

- Fig. 7: Personally I think the ferret labels should be removed from the graphic. You can acknowledge use of the ferret program in the acknowledgements.

- Fig. 8: Not the impact on run time, but the run time itself for different numbers of nodes is shown in the figure. I assume that all the tests are performed without any output. Could you mention this somewhere (e.g. in the setup description to be added to this article?) How exactly do you deduce the time for MESSy? Do you assume GPC is MESSy (including MECCA)-time and the rest is ECHAM time?

- Fig. 11: Its MESSy not MESSY (2x)

[Figure]

---

## Referee Comment (RC2) · Anonymous Referee #2 · 3 May 2016

The article adresses the important issue of the performance of Earth System Models on future heterogeneous cluster architecture. Within the DEEP project the GCCM EMAC has been ported to a cluster booster system.

Initially, the authors provide a performance analysis to identify the dominant factors in computational workload and to pin down the bottlenecks that prevent scalability. They conclude that with respect to performance issues the choosen EMAC setup could be split into two parts. The KPP based chemistry integrator MECCA as the major contributor to the computational workload is well suited for massive parallel treatment and the candidate for the booster architecture. Whereas the rest has limited scalability due to strong coupling and high communication demands and remains on the cluster.

The authors briefly present the refactoring that is needed to remedy the issues preventing the code to run successfully on the cluster booster system. With this changes the model shows impressively enhanced performance and scalability. In a nutshell the article provides recipes to successfully port ESMs on the future heterogeneous architecture. This is an important result that definitely deserves to be published.

Nonetheless, I require major revisions. There is a considerable lack in detail, important information is missing. The reader simply can't really reconstruct what the authors have done, the results are not reproducable. The issuses of the code refactoring are just presented as keywords. The reader does not really know what has been done. Every code refactoring topic has to be exemplified. The authors should present the old structure (not the whole code, but the essential programm structure that needs to be modified) in comparison with the new logic. This demands a considerable rewriting of the article.

In summary, I strongly support to publish this article. But major revisions are required to assure reproducibility and to render it really beneficial for the modelling community.

Specific Comments

2.1 Phases

-page 3, lines 9-15 : Redundant, you have said that already in the introduction.

-page 3, lines 26-27 : "Furthermore, even a coarser..." What does this sentence mean?

2.2 Dominant factors

Fig. 2 is not mentioned.

-page 4, line 8 : "data size scales with the the square of model resolution" This is only true, if the vertical resolution is unchanged.

-page 4, lines 11-18: The discription of Fig. 4 is very confusing and maybe wrong. Doesn't it start with the transformation from spectral space into Eulerian space, followed by grid point calculations and ending with transformation back into spectral space?

-page 4, lines 23-26: "Model performance depends largely on a virtual ..." What do these sentences mean? If the model performance largely depends on that, this should be explained better.

-page 5, line 3 : Fig. 5 is not really needed. The numbers could be mentioned in the text.

2.3 Scalability considerations

In general, chapters 2.2 and 2.3 deal with the same topic. Maybe just one chapter is needed.

-page 6, line 9 : It isn't clear how Fig.7 is made. Does it show the integrated computation time of a column?

-page 6, lines 10-15 : Has been said before.

-page 6, lines 16-20 : Fig. 8, what is the configuration of MareNostrum 3? What is the number of MPI-processes on a node?

3.1 Intranode taskification

Doesn't Fig. 10 contradict with Fig. 9 ? ECHAM in EMAC is not affected by OmpSs and with 32 Nodes 40% of CPU time is with ECHAM. This amount strongly increases with the number of nodes. Why does Fig. 10 not reflect this?

-page 7, lines 5-6 : Refactored, but how? Please exemplify.

3.2 Internode taskification

Please exemplify. What has been changed in the code? Please provide examples from the code. The reader can't really reproduce or reconstruct what you have done.

3.3 Vectorisation

-page 8, lines 21-27 : Redundant, has been said before.

-page 8 line 28 - page 9, line 5 : What have you done? Have you changed the original code? Please exemplify.

Fig. 11 is not really necessary, the numbers are already in the text.

-page 9, lines 11-13 : What does this mean? How has the code been changed? Please exemplify.

4 Attainable performance

-page 9, lines 18-29 : This is very confusing. How is this done? Please provide equations.

5 Conclusions

-page 11, lines 4-7 : How is this done? This has to be exemplified in the article.

Code Availability Is the recoded EMAC from the DEEP project available to the public?

––––––––––––––––––––––––––––––

---

## Author Comment (AC1) · 30 May 2016

Dear Editor,

We would like to thank the referees for their careful reading of our manuscript, gmd-2015-262, " Earth System Modelling on System-level Heterogeneous Architectures: EMAC (version 2.42) on the Dynamical Exascale Entry Platform (DEEP)".

We are most grateful for the comments, constructive criticism and very useful suggestions received on how to improve the paper.

Please find attached our detailed answers to the questions and a new version of the paper, which we hope satisfactorily address the points raised during the discussion. In particular, we have extensively expanded the description of code refactoring, implementation and technical developments.

**Anonymous Referee #1**

The article reports about one part of the DEEP project examining the approach of heterogeneous cluster-computing for Earth System models. In particular, this article is about the GCCM EMAC. A performance analysis shows, that the chemistry submodel MECCA is the bottleneck of usual EMAC simulations. As MECCA is solving the chemistry within each grid box independently, it is embarrassing parallel and thus the ideal candidate for application on a Booster architecture.

The article addresses a very important issue of Earth system modelling. The fact that these codes have a long history brings about that they are not at all optimised for modern computer architectures. Thus the refactoring of the code, to used the possibilities provided by current and future computing architectures is a very important issue.

Nevertheless, there are some issues that need to be improved upon revision. First of all the specific model configuration used for the scaling tests is not provided. But the setup heavily influences the performance of the model. Additionally, the explanations provided in this article are only correct for certain model setups. Therefore the setup needs to be provided. Secondly, this is a GMD article and the authors classified it "Development and technical paper". Thus I expect the authors to provide much more details about the developments themselves. How exactly have they been implemented? In general, the performance analysis is as long as the description of the developments. The latter should be the main part of the article from my point of view.

Below I give the details about the points raised above and name some additional issues. In summary, I am very much in favour of publishing this article. Nevertheless, major revisions are required.

Specific Comments

1. The authors often use the term "meteorological model" when they mean the dynamical core of the model. Meteorology comprises the dynamical processes as well as the physical processes such as cloud and precipitation formation or radiation. EMAC uses the dynamical core of the ECHAM model, but all physical processes are – by now – modularised as MESSy submodels. In ECHAM the dynamical core operates in spectral space, the physical processes are implemented in grid point space. All MESSy submodels (so far) are implemented as additions in the grid point space. Therefore the authors should be consistent in the terms they use throughout the article. Please change accordingly: • page 2 line 12 • p.3, l. 2 + l. 6 • p.5 l.24 • p. 10 l.10 • Figure 2, left box This list is not complete. So please check throughout the article.
   **Corrected throughout.**
2. To really understand the performance analysis, the full EMAC setup should be listed somewhere. If an ESCiMo-Setup or another published setup was used, this could simply be

cited. If not, I would prefer to have a description, containing all details relevant for the publication, in the appendix. A zip-file containing the full namelist setup should be provided in the supplement.

**Detailed description of the model setup, including namelist description is now provided in Sec. 2.2.**

3. Connected to the previous point: You characterize MECCA as the submodel computing "the chemical kinetics of the homogeneous gas-phase chemistry of the atmosphere," ... (e.g., p.3 l.13; p.4 l.29)
In all MESSy setups which are not only focussed on the lower troposphere, heterogeneous processes on ice cloud are included via reaction rates provided by MSBM. As I do not know the specific setup of this study, I can not judge, if this statement is correct with repsect to it. As a general statement it is definitely not correct.
**Changed all instances to be exact.**

4. If MECCA is employed with a pure gas phase mechanism in this test setup, what will be the result, if heterogeneous reactions are included? This would establish a second source of imbalance caused by the appearance of PSCs.
**The second source of imbalance by heterogeneous reactions is also automatically alleviated by the dynamical load balance using the massive parallelisation in the Booster. Our proposed solution is agnostic to the specific origin of load imbalance in the chemistry calculation. Added to text.**

5. SCAV uses KPP as well. In maximum it is called four times during one time step (for grid/subgrid scale liquid/ice clouds). Here the load imbalance is caused by the distribution of clouds over the model domain and additionally, SCAV is column bound. At least in the conclusion or outlook I'd like to see a statement, how easily your developements could be applied to SCAV and (maybe) if you expect performance gains for SCAV as well.
**In principle, the MECCA implementation is directly applicable to the case of SCAV. The actual performance gain would be dependent on the exact setup and remains to be tested. Added relevant statement in the conclusion.**

6. Chapter 3: Model developments What I am really missing in this chapter is the "development" i.e., an explicit mentioning of the code changes required. You describe them superficially with some words, what - at least for a MESSy developer / user - is really interesting is what the code changes look like. For which of the described changes did you change which code parts? Do they require changes in the MESSy submodel interface layer (SMIL) or in the core layer (SMCL). Here it is most interesting, if these developments require a change in the automatically generated code. If yes, MECCA knows two stages of "automation". First KPP produces the code automatically from the equation file. Secondly, KP4 can be applied in order to remove the indirect indexing and expand the original KPP code by an additional dimension to enable a better performance due to better cache usage. If there are changes in the automatically generated code parts, did you change the scripts performing the automation or did you just chance one MECCA setup (and running xmecca once would destroy all you efforts)?.
I assume that this information is only of interest for MESSy or MECCA developers / users and not to the general readership. Therefore I recomment to add the most important information to the article itself and to add a supplement describing the changes in more detail and providing information about how to use it (something like a user manual), which also is in accordance to the GMD guidelines. Additionally, a description, how the optimise the setup as described in p.8 ll.10-12 should be provided.
**All changes are in the MESSy MECCA KPP source and no additional changes are needed in the the MESSy submodel interface layer (SMIL) or in the core layer (SMCL). Since the source is**

**automatically generated, the changes have to propagated to the generator after the KP4 mechanism runs (applied in order to remove the indirect indexing and expand the original KPP code by an additional dimension to enable a better performance due to better cache usage). The propagation is still work in progress (also to enable GPGPU usage) and falls outside the scope of this manuscript. Added to the text. To use the code as is, one just needs to compile with the new messy_mecca_kpp.f90 source file in place, so no specific instructions are required.**

**The compiler directives as implemented as pragmas (comments in the source), controlled with definitions at compile time automatically (by invoking only if SMP/Mercurium are present), thus no additional user input is required.**

**Finally, added clarification how the performance can be optimised based on the shared memory of each architecture. This can also be performed empirically by scaling tests with varying runtime parameter.**

**All of the above has been added to the text in Sec. 3.**

7.  p. 8 ll. 1-19: It is not clear to me, what these paragraphs are about: ll.1-6 are a repetition of what was written earlier. ll. 7-12: claim that the code can be optimised tailormade for each architecture but does not tell how. ll.13-19 simply state that these changes increase the performance, but does not show any proof. Somehow I miss the point here...
    **Several lines removed to avoid repetition. Added clarification how the performance can be optimised based on the shared memory of each architecture. This would enable the current implementation to run on future Intel MIC architectures with different memory amounts. Added information to the text.**

8.  Section 3.3 In this section the authors must be much more precise. It is not clear, what exactly the authors are discussing here. Principially, the ECHAM (and thus the MESSy) grid point code is decomposed in a way of artificial latitude bands. ECHAM provides a so-called "local loop": here a loop over the second horizontal dimension is established, reducing the size of the fields forwarded to the individual submodels within the local loop by one horizontal dimension. The length of the remaining horizontal dimension is called vector length and can be chosen by namelist for optimisation on different computing achitectures. Thus MECCA, which is called in the local loop is called with this one horizontal dimension only. For a better performance the original KPP output can be expanded by this additional horizontal dimension times the vertical dimension. This code is automatically produced by calling KP4. It is not clear to me, about which of these different aspects the authors talk exactly. So please clarify this issue.
    **Section 3.3 was removed and relevant information was merged with Sections 3.1 and 3.2 to make more clear and keep the text precise.**

9.  Code Availability: You state the general terms for the MESSy code. Nevertheless, you are presenting new code developments, therefore it would be good to know, if it will become part of the official MESSy version soon and if it is possible to get hold of the code prior to this.

    **Code developments are not presently available in the official MESSy distribution. The current implementation is specific to the pre-production DEEP system, hardware and software stack. Added "Changes the source code to implement system-level heterogeneous offloading are currently specific to the Mercurium compiler, Nanos++ runtime and proprietary Parastation MPI and are hosted on the DEEP project version control repository. "**

Minor issues

• paragraph p.3 ll.22-27, it would be good to have examples here. You provide them on page 5 last two lines, but they are already here useful to understand which kind of processes you are talking about.

**Added examples also at p.3.**

• I have mixed feelings about Fig. 1. It looks nice, but is it really required? Additionally, as processes as deposition occur on the right hand side, this figure is in contradiction to Fig. 2.

**Agreed. This was also commented on by the other Referee and Fig. 1 was removed.**

• p.4, l.1: Shouldn't it be Figure 2?

**Corrected.**

• p.4, ll.8-10: it scales with the square of the horizontal resolution.

**Added "horizontal".**

• p.5 l.13/ Table 1: Table 1 does not give a clue about the model setup, it only contains different numbers calculated from the resolution of the model. In my opinion the first column does not contain any information I would expect in a table. That the numbers indicate the possibility for strong scaling should be stated in the text not in the table itself. What is meant by "42 coefficients"? Which coefficients? Do you mean because of T42? Than the reader anyhow understands what T42 refers to, or he/she does not become any wiser by reading "42 coefficients" (my opinion). The table would be much better readable, if columns and rows would be switched

**Removed the first row. Transposed rows with columns. Specified in the text that this relates to strong scaling. Removed "42 coefficients"**

• Table 2: Here the last statement for table 1 applies even more: the table is much better readable, if switch columns and rows are switched.

**Transposed rows, columns of Table 2.**

• p.6 ll. 3-6: This is not fully correct. From your description I visualise a decomposition where the domain is split up in rectangular grid boxes. But ECHAM is using a decomposition where each task gets two (independent) latitudinal bands (of arbitrary length in the longitudinal range). These bands are usually not even adjacent to each other.

**ECHAM employs a load-balancing distribution for the imbalance between the Earth's hemispheres, limited to two rectangular sets of grid points distributed symmetrically to the equator. While it is common and computationally beneficial to extend them along latitudes, ECHAM allows for square of latitudinally extended decompositions. This has been added to the text.**

• p.6 l. 8: add location of natural and anthropogenic emissions.

**Added emissions origin.**

• p.6 ll. 16-19: Why are you only describing the results for ECHAM seen in Fig. 8? Add something about "computing time for MESSy is still decreasing" and "computing time for MECCA decreases stronger than MESSy". Additionally, please point to the logarithmic scale and to the fact, that "MESSy" means without MECCA.

**Added required changes to text.**

• p.6 ll.21-23: I am missing a conclusion, you just describe the plot.

**Added conclusion.**

• p.7 last line: "each time step", "step" is missing.

**Added.**

• p.8 l.31: What is meant by "domain-specific language" ?

**A domain-specific language (DSL) is a computer language specialized to a particular application domain. This is in contrast to a general-purpose language (GPL), which is broadly applicable across domains, and lacks specialized features for a particular domain. Nonetheless, this section has been removed in agreement with other referee comment.**

• p.9 l.29 ; It would be helpful for the reader if you could mention the colours of the respective lines in the text. Do I assume right that "MPI" is the sum of ECHAM+MESSy and MECCA? First I thought it is the MPI communiction time. Please try to clarify your description.

**Added references to line colours and clarified description.**

• Fig. 1: For me this figure produces more questions instead of assisting in understand the distribution on the two different computing architecture parts. The "base model cluster" part contains a picture of a cloud, i.e., physical processes, and the "atmospheric chemistry booster" part does not only contain the chemical mechanism, but also deposition, thus it is not quite clear where the separation between cluster and booster should appear.
**Removed Fig. 1.**

• Fig. 3: It is not clear on which ground the colours of the boxes are chosen. Personally I think, the figure overemphasises the dynamical core (including the transformations from grid point to spectral and vice versa). Because the grid point calculations contain much more sub processes which are completely left out by this figure.

**Removed Figure 3 altogether to avoid overemphasising the dynamical core.**

• Fig. 5: Please provide a more descriptive caption for this figure. Not every reader is familiar with Scalasca output.
**Figure removed - added the numbers in the text.**

• Fig. 7: Personally I think the ferret labels should be removed from the graphic. You can acknowledge use of the ferret program in the acknowledgements.
**Removed labels to improve figure.**

• Fig. 8: Not the impact on run time, but the run time itself for different numbers of nodes is shown in the figure. I assume that all the tests are performed without any output. Could you mention this somewhere (e.g. in the setup description to be added to this article?) How exactly do you deduce the time for MESSy? Do you assume GPC is MESSy (including MECCA)-time and the rest is ECHAM time?
**Added in the text that all tests are performed with no output. Updated Fig. 8 and caption to "run time", instead of "impact on run time". MESSy is counted as all GPC run time that is not in ECHAM dynamical calculations or MECCA chemistry processes.**

• Fig. 11: Its MESSy not MESSY (2x)
**Fig. 11 removed, as suggested by Referee #2. Numbers in text.**

The article adresses the important issue of the performance of Earth System Models on future heterogeneous cluster architecture. Within the DEEP project the GCCM EMAC has been ported to a cluster booster system. Initially, the authors provide a performance analysis to identify the dominant factors in computational workload and to pin down the bottlenecks that prevent scalability. They conclude that with respect to performance issues the choosen EMAC setup could be split into two parts. The KPP based chemistry integrator MECCA as the major contributor to the computational workload is well suited for massive parallel treatment and the candidate for the booster architecture. Whereas the rest has limited scalability due to strong coupling and high communication demands and remains on the cluster.

The authors briefly present the refactoring that is needed to remedy the issues preventing the code to run successfully on the cluster booster system. With this changes the model shows impressively enhanced performance and scalability. In a nutshell the article provides recipes to successfully port ESMs on the future heterogeneous architecture. This is an important result that definitely deserves to be published.

Nonetheless, I require major revisions. There is a considerable lack in detail, important information is missing. The reader simply can't really reconstruct what the authors have done, the results are not reproducable. The isssues of the code refactoring are just presented as keywords. The reader does not really know what has been done. Every code refactoring topic has to be exemplified. The authors should present the old structure (not the whole code, but the essential programm structure that needs to be modified) in comparison with the new logic. This demands a considerable rewriting of the article.

In summary, I strongly support to publish this article. But major revisions are required to assure reproducibility and to render it really beneficial for the modelling community.

Specific Comments

2.1 Phases
-page 3, lines 9-15 : Redundant, you have said that already in the introduction.

**Removed.**
-page 3, lines 26-27 : "Furthermore, even a coarser..." What does this sentence mean?

**Removed as it was indeed not clear and deemed unnecessary.**

2.2 Dominant factors
Fig. 2 is not mentioned.
**Corrected reference to Fig.2 previously in the text.**
-page 4, line 8 : "data size scales with the the square of model resolution" This is only true, if the vertical resolution is unchanged.
**Specified "horizontal" resolution**.
-page 4, lines 11-18: The discription of Fig. 4 is very confusing and maybe wrong. Doesn't it start with the transformation from spectral space into Eulerian space, followed by grid point calculations and ending with transformation back into spectral space?
**Changed from model time step to one cycle to be clear and reformatted – Fig.4 starts at the end of a model timestep and not the beginning.**

-page 4, lines 23-26: "Model performance depends largely on a virtual ..." What do these sentences mean? If the model performance largely depends on that, this should be explained better.
**Removed to make text more clear.**

-page 5, line 3 : Fig. 5 is not really needed. The numbers could be mentioned in the text.
**Removed figure. Added numbers in the text.**

2.3 Scalability considerations In general, chapters 2.2 and 2.3 deal with the same topic. Maybe just one chapter is needed.
**Consolidated into a single subsection.**

-page 6, line 9 : It isn't clear how Fig.7 is made. Does it show the integrated computation time of a column?

**Fig 7 shows the column maximum for a single time-step. Clarified in the caption.**
-page 6, lines 10-15 : Has been said before.
**Removed.**
-page 6, lines 16-20 : Fig. 8, what is the configuration of MareNostrum 3? What is the number of MPI-processes on a node?
**Added configuration of Marenostrum 3.**

3.1 Intranode taskification

Doesn't Fig. 10 contradict with Fig. 9 ? ECHAM in EMAC is not affected by OmpSs and with 32 Nodes 40% of CPU time is with ECHAM. This amount strongly increases with the number of nodes. Why does Fig. 10 not reflect this?

**ECHAM takes a higher percentage of the time not only because ECHAM runtime increases but alos because the other components still scale. This is more readily apparent in Fig. 8 which shows absolute time taken (rather than percentage).**

-page 7, lines 5-6 : Refactored, but how? Please exemplify.
**Exemplified in detail in the text. Please see reply to Referee #1 major comments 6&7.**

3.2 Internode taskification
Please exemplify. What has been changed in the code? Please provide examples from the code. The reader can't really reproduce or reconstruct what you have done.
**Exemplified in detail in the text. Added code examples of the refactoring and required compiler directives to aid reproduction/reconstruction.**
**Please see reply to Referee #1 major comments 6&7.**

3.3 Vectorisation

-page 8, lines 21-27 : Redundant, has been said before.
**Removed.**

-page 8 line 28 - page 9, line 5 : What have you done? Have you changed the original code? Please exemplify. Fig. 11 is not really necessary, the numbers are already in the text.

**Removed Fig.11. Code modifications are now discussed in detail in the text.**

-page 9, lines 11-13 : What does this mean? How has the code been changed? Please exemplify.
**Sec. 3.3 is now removed and required information merged with Sections 3.1 and 3.2 with detailed information on changes to exemplify.**

4 Attainable performance

-page 9, lines 18-29 : This is very confusing. How is this done? Please provide equations.
**Added equations, better description and highlighted Figure line colours in the description in the text to make more clear.**

5 Conclusions

-page 11, lines 4-7 : How is this done? This has to be exemplified in the article.
**The load imbalance observed in MECCA will be automatically alleviated through dynamic load balancing by minimising the individual task size to one grid box and decoupling the model-domain location from the task execution on the physical CPU and transferring it to any available core on the Booster. Added to the text.**

Code Availability
 Is the recoded EMAC from the DEEP project available to the public?

**Added to code availability that "Changes to the source code to implement system-level heterogeneous offloading are currently specific to the Mercurium compiler, Nanos++ runtime and proprietary Parastation MPI and are hosted on the DEEP project version control repository."**

---

## Referee Report (RR1)

The authors included all the required additional information into the article. Therefore I am much in favour of publishing this article. Some very minor issues remain:

- page 5, line 2: abbreviation JSC not introduced
- page 6 line 20: full stop after bracket missing
- page 6 l.30 / page 7 l. 1 and p. 7, l. 23/24 : What is "runtime" as it seems to be able to do something (from your sentences). Do you mean "at runtime" or "during runtime" or do you mean by runtime something I do not understand?
- p.7 l. 8: "All changes are in the MESSy MECCA KPP source" → " All changes are in the automatically generated MESSy MECCA KPP source code "..
- p. 8, l. 2: Fig. 4 → Fig. 7
- p. 9, l. 9: NBL is not introduced here. (at least add "(see below)" behind NBL.
- p. 11, l. 19: Fig. 5 → Fig. 8
- p. 12, l. 8: Fig. 5 → Fig. 8
- p. 13, l. 19: SCAV is not the aerosol submodel (that would be GMXe, also dealing with KPP), SCAV calculated scavenging ...
- p. 13, l. 19: remove "also"